# Predictive value of hemogram parameters in malignant transformation of the endometrium in patients with different risk factors

**Aysun Firat**[1]*, **Aysegul Ercan**[1], **Cengiz Mordeniz**[2], **Fatma Ferda Verit Atmaca**[1]

**1** Department of Obstetrics and Gynecology, Istanbul Education and Research Hospital, University of Health Sciences Turkey, Istanbul, Turkey, **2** Department of Anesthesiology and Intensive Care, Tekirdag Namik Kemal University, Tekirdag, Turkey

\* aysunfiratsbuieah@gmail.com

## Abstract

### Objectives

To investigate whether the pretreatment hemogram parameters and their ratios can be used in predicting the endometrial transformation in patients with abnormal uterine bleeding.

### Material and methods

Records of all patients who underwent an endometrial histopathological evaluation between 2011 and 2021 were investigated. Hemogram, neutrophil to lymphocyte ratio (NLR) and platelet to lymphocyte ratio (PLR) were analyzed. Chi square and Mann Whitney U tests were used for analysis. P<0.05 was considered statistically significant.

### Results

427 patients were included, of whom 117 were presented with endometrial hyperplasia without atypia (27.4%; mean age, 42±9.7; Group II), 70 with atypia (16.3%; mean age, 53.4±9; Group III), 102 with early endometrial cancer (EC) (23.8%; mean age, 63±7.8; Group IV) and 38 with advanced disease (8.8%; mean age, 63.3±10.5; Group V). Patients without pathology constituted the control group (23.4%; mean age, 42.2±9.5; Group I). Risk factors for atypia and carcinoma were determined as age, postmenopausal state, obesity, diabetes, and increased estrogen exposure (each, p<0.05). There was no significant difference in NLR and PLR (p>0.05). However, hemoglobin and hematocrit levels were higher in Groups IV and V (13.9 vs 13.1 mg/dL, and 39.1 vs 38.8%, respectively; p<0.01). Platelet value was significantly higher in Groups III to V (282x10$^9$/L, 283x10$^9$/L and 295x10$^9$/L; p<0.05, p<0.05 and p<0.01, respectively).

**Data Availability Statement:** Data from this study are available upon request since there are legal

restrictions (by Turkish Ministry of Health) on sharing data publicly. However, we anonymized the patients' identities and protocol numbers on the system and saved whole data in Excel form. Data protection process and audit are supplied by Local Ethics' Committee. The formal email and phone and the name of nonauthor secretary of Ethics Committee; Selda KAYALI Secretary; Kıymet GÜLER Vice secretary; GENEL 0 (212) 459 6220 ieahetikkurul@gmail.com / istanbuleahpersonel@gmail.com; https:// istanbuleah.saglik.gov.tr/TR-168206/kurul-uyeleri. html). The first and the corresponding author of the study (Aysun Firat, MD) may also send them by email (aysunfiratsbuieah@gmail.com) on request.

**Funding:** The authors received no specific funding for this work.

**Competing interests:** The authors have declared that no competing interests exist.

## Conclusions

Our findings support the impact of inflammation on malign transformation from normal endometrial mucosa to atypia and carcinoma. NLR and PLR values showed no statistical difference. Instead, thrombocytosis may have a predictive role in EC.

## Introduction

Endometrial cancer (EC) is the most common gynaecological malignancy in developed countries, and endometrial hyperplasia is its precursor [1]. The most common presentation is abnormal uterine bleeding [2]. Estrogen, unopposed by progesterone, stimulates endometrial cell growth, and well-known risk factors reflecting this etiology are obesity with excessive peripheral conversion of androgens to estrogen, anovulation associated with perimenopausal period or polycystic ovary syndrome (PCOS), estrogen-secreting ovarian tumours, excessive menstruation, early menarche, late menopause, the use of hormone replacement therapy (HRT) or tamoxifen [1–3]. Postmenopausal elderly and diabetic women have also higher risk in developing EC [4, 5].

Chronic inflammation is the common feature of all these risk factors, and it may play an important role in the transition from normal endometrium to hyperplasia, polyp formation, atypia and malignancy [6]. Obesity and diabetes mellitus (DM) promote subclinical chronic inflammation, and menstruation itself mimics an inflammatory process with its proliferative, secretory and menstrual phases [5, 7]. Chronic inflammation is mitogenic, and it increases the possibility of replication error and causes ineffective DNA repair [8].

A precancerous or cancer-related inflammatory microenvironment can be reflected in the blood as measurable parameters impacting immunogenicity, such as neutrophilia, thrombocytosis, and lymphopenia [9]. Recent researches have focused on these inflammatory indicators, and owing to the challenge related to the clinical application of separate counts of them, ratios have been used as predictive factors [10, 11]. Current studies have demonstrated that neutrophil-to-lymphocyte ratio (NLR) and the platelet-to-lymphocyte ratio (PLR) are important prognostic indicators in a variety of solid tumors, including EC [10–14]. However, the results of these studies are still inconclusive. Besides these parameters, cancer-associated thrombosis and erythropoiesis were known long ago, and could be checked with a simple hemogram [11, 15].

Several biomarkers associated with endometrial hyperplasia and the subsequent carcinoma have been investigated [16]. However, none of them predicts the disease accurately enough to be clinically useful. Understanding of the premalignant stages of EC development may enable an earlier diagnosis, and may facilitate an appropriate risk stratification to take timely and appropriate therapeutic actions. An easy, reproducible and simple marker is still needed to predict the transition to invasive carcinoma, and distinguish this from the benign lesions.

The purpose of this study is to investigate whether the pretreatment hemogram parameters, and NLR and/or PLR can be used in predicting the stage of endometrial transition in patients with abnormal uterine bleeding.

## Material and methods

Patients provided informed written consent to have data from their medical records used in research. The study was approved by our institutions's Ethics' Committee (SBU-IEAH/21.05.2021/2837). Medical records of patients who were admitted with abnormal uterine

bleeding and underwent an endometrial histopathological evaluation between 2011 and 2021 were reviewed. Pathological findings reporting thick endometrium were further investigated for precancerous lesions such as hyperplasia and polyp, atypia, and for EC. The results of blood samples just before the curettage procedure were recorded at Excel programme (Microsoft 2017, Illionis, US). Patients with missing information in their files and those with any documented infectious or inflammatory disorders were excluded.

Demographics and risk factors for endometrial pathology were recorded. Risk factors were determined through an extensive PubMed search including last two decades (2000–2020), and documented as: older age, high body mass index (>25), increased estrogen exposure (early menarche, late menapouse, heavy menstruation, nulliparity, endometriosis, HRT and tamoxifen use, PCOS) and postmenopausal disease.

Patients were investigated in 5 groups: Group I (Control group; patients without endometrial pathology), Group II (patients with benign endometrial hyperplasia or polyp), Group III (atypic endometrial hyperplasia), and Groups IV and V (EC). Patients with a diagnosis of EC were further staged according to the International Federation of Gynecology and Obstetrics (FIGO) guidelines as early (stage 1 and 2; Group IV) and advanced disease (stage 3 and 4; Group V). Main hemogram parameters were hematocrit (percentage, %), hemoglobin (g/dL), leukocyte (x$10^9$/L), neutrophil (x$10^9$/L), lymphocyte (x$10^9$/L) and platelet (x$10^9$/L). Red cell distribution width (RDW), as a red blood cell parameter that measures variability of red cell volume/size (anisocytosis), was also reported statistically as coefficient of variation (CV, in %) and standard deviation (SD, in fL) (RDW-CV and RDW-SD, respectively).

NLR was defined as the absolute neutrophil count divided by the absolute lymphocyte count, and PLR was defined as the absolute platelet count divided by the absolute lymphocyte count. The ratios and other hemogram parameters were correlated with histopathological findings and clinical risk factors.

Statistical package for social sciences (SPSS version 11.5) was used for the statistical analysis. Descriptive values were expressed as number (n), %, median or mean with standard deviation (SD).Chi-square, Student's t and Mann-Whitney U tests were used for nominal and categorical values, and Kruskal–Wallis test was used to compare the nonparametric variables. Group-comparisons were done with one-way ANOVA and Tukey HSD tests to ascertain the group that cause the difference. P<0.05 was considered statistically significant.

## Results

A total of 427 female patients (mean age±SD, 49.8±12.5) were included, of whom 117 were presented with endometrial polyp or hyperplasia without atypia (27.4%; mean age, 42±9.7; Group II), 70 with atypia (16.3%; mean age, 53.4±9; Group III), 102 with early stage endometrial cancer (23.8%; mean age, 63±7.8; Group IV), and 38 with advanced/metastatic carcinoma (8.8%; mean age, 63.3±10.5; Group V). Patients without significant pathology constituted the control group (23.4%; mean age, 42.2±9.5; Group I).

The age difference between the groups was significantly higher in Groups IV and V than the other groups (p<0.01, Table 1). The mean age in patients with EC (Groups IV and V) was 63 years. Patients with endometrial atypia (Group III) were also older in comparison to the patients with normal endometrium (Group I) and the patients with endometrial hyperplasia or polyps (Group II) (53.4 years vs 42.2 and 42 years; each, p<0.05). Most of the patients in Groups III, IV and V were postmenopausal (67%, 83% and 84%; p<005, p<0.001 and p<0.001, respectively), and there was a positive correlation with age. Other than age and postmenopausal status, the risk factors for atypia and EC were determined as BMI (mean 29 in Groups III and V; p<0.05 vs mean 32 in Group IV; p<0.01, respectively), DM (8% in Group

**Table 1. Demographics and risk factors.**

| | Group I (n = 100) | Group II (n = 117) | Group III (n = 70) | Group IV (n = 102) | Group V (n = 38) |
|---|---|---|---|---|---|
| Mean age (year)±SD (standard deviation) | 42.27±9.50 | 42.05±9.78 | 53.41±9.04* | 63.07±7.84** | 63.34±10.56** |
| BMI (body mass index) | 26.40±7.44 | 26.05±6.32 | 29.63±5.90* | 32.80±5.89** | 29.98±5.42* |
| DM (diabetes mellitus) | 2 (2%) | 4 (3.41%) | 3 (4.28%) | 9 (8.82%)* | 4 (10.5%)* |
| Increased estrogen exposure | 17 (17%) | 18 (15.38%) | 17 (24.28%)* | 25 (24.50%)* | 12 (31.57%)** |
| Premenopausal disease | 76 (76%)** | 87 (74.35%)** | 23 (32.85%)* | 17 (16.66%) | 6 (15.78%) |
| Postmenopausal disease | 24 (24%) | 30 (25.64%) | 47 (67.14%)* | 85 (83.33%)** | 32 (84.21%)** |

Group I: control group, Group II: benign endometrial hyperplasia, Group III: atypia, Group IV: early endometrial cancer, Group V: advanced endometrial cancer, BMI (18–24: normal, 25–29: overweight, 30–34: obese, >34: morbid obese), increased estrogen exposure (early menarche, late menapouse, heavy menstruation, nulliparity, endometriosis, HRT: hormon replacement treatment, PCOS: polycystic ovary syndrome)

*p<0.05: statistically significant

**p<0.01: statistically very significant

IV, and 10% in Group V, respectively; each, p<0.05) and increased estrogen exposure (>24% of the patients in Groups III, IV and V; each, p<0.05).

There was no statistically significant difference between the groups in RDW-CV, RDW-SD, leukocyte, neutrophil and lymphocyte values (each, p>0.05; Table 2). The groups were compared by calculated NLR and PLR results, and there was no statistically meaningful relation, as well (each, p>0.05). However, hemoglobin and hematocrit levels were higher in Groups IV and V (13.9 mg/dL vs 13.1 mg/dL, and 39.1% vs 38.8%, respectively; each, p<0.01; Table 2). Similarly, platelet value was significantly higher in Groups III, IV and V (282x10$^9$/L, 283x10$^9$/L and 295x10$^9$/L; p<0.05, p<0.05 and p<0.01, respectively).

## Discussion

The link between cancer and inflammation was first suggested by Virchow in the 19th century, and since then, inflammation is known to be both a cause and a consequence of cancer [6, 17].

**Table 2. Hemogram parameters in groups.**

| | Group I (n = 100) | Group II (n = 117) | Group III (n = 70) | Group IV (n = 102) | Group V (n = 38) |
|---|---|---|---|---|---|
| Hematocrit (%) | 36.11±4.52 | 36.11±4.66 | 37.35±3.92 | 39.10±4.39** | 38.88±5.33** |
| Hemoglobin (g/dL) | 11.83±1.70 | 11.81±1.99 | 12.35±1.57 | 13.90±1.60** | 13.10±2.09** |
| Leukocyte (x10$^9$/L) | 8.28±3.76 | 8,20±3.2 3 | 7.52±2.38 | 8.19±2.27 | 8.26±2.29 |
| Neutrophil (x10$^9$/L) | 5.40±3.57 | 5.52±3.14 | 4.73±2.11 | 5.08±1.85 | 5.44±2.00 |
| Lymphocyte (x10$^9$/L) | 2.17±0.91 | 2.08±0.73 | 2.54±0.70 | 2.57±2.33 | 2.88±0.71 |
| Platelet (x10$^9$/L) | 251.28±67.40 | 256.75±69.62 | 282.51±64.56* | 283.38±70.68* | 295.15±75.02** |
| RDW-CV (%) | 15.30±4.15 | 14.81±3.27 | 14.36±2.15 | 13.87±1.55 | 14.52±2.36 |
| RDW-SD (fL) | 43.76±9.79 | 42.57±7.37 | 41.48±3.70 | 42.08±3.72 | 43.39±4.42 |
| NLR | 2.49±5.17 | 2.52±4.23 | 2.27±1.89 | 2.23±1.48 | 2.61±1.72 |
| PLR | 126.62±75.98 | 126.52±70.46 | 128.32±52.78 | 128.99±55.00 | 129.18±78.60 |

Group I: control group, Group II: benign endometrial hyperplasia, Group III: atypia, Group IV: early endometrial cancer, Group V: advanced endometrial cancer, RDW-CV: red blood cell distribution width-coefficient of variation, RDW-SD: red blood cell distribution width-standard deviavtion, HBRDW-SD: hemoglobin blood cell distribution width-standard deviavtion, NLR: neutrophil/lymphocyte ratio, PLR: platelet/lymphocyte ratio

*p<0.05: statistically significant

**p<0.01: statistically very significant

Menstruation itself, during which the endometrium goes through proliferative, secretory, and menstrual phases, is also an inflammatory process, and it is associated with activation of cytokines resulting in the shedding of inflammatory cells [18]. Proinflammatory microenvironment increases the estrogen production, and estrogen directly regulates the production of inflammatory cytokines, growth factors and their corresponding receptors, as well [19]. Therefore, inflammation works in conjunction with estrogen exposure in the transition of normal endometrium to atypic hyperplasia and carcinoma. Risk factorsfor ECassociated with unopposed estrogen exposure areearly menarche, late menopause, heavy menstruation, PCOS, HRT and tamoxifen use etc., and all are known as factors increasing exposure of the endometrium to inflammation [2–4]. Atypical endometrial hyperplasia is usually seen in older women, but it may also develop in younger women who do not ovulate or are obese [20]. Local and systemic effects of obesity (BMI>29) and DM are other well-known promoters of subclinical chronic inflammation [5, 20].

In line with the current literature, the risk factors for atypia and EC, determined in our study, were older age (mean 53 years in atypia and 63 in EC vs 42 years in patients with normal mucosa or benign endometrial hyperplasia), higher BMI (>29 in both atypia and EC), DM (8% in early EC and 10% in advanced disease vs 2 to 4% in other patients), and increased estrogen exposure (31% in advanced EC vs 24% in atypia). Over 80% of the patients with EC and over 60% of the patients with atypia were postmenopausal. All of the determined risk factors in the present series seem to be associated with subclinical chronic inflammatory condition or disease, such as getting older or fat, DM, etc. Our findings also support the theory suggesting the impact of chronic inflammation on malign transformation from normal endometrial mucosa to atypia and carcinoma.

Recently, the hematological markers of inflammation have been found to be potentially useful in determining the stage or prognosisof a variety of tumors, including soft tissue sarcomas, colorectal cancer, breast carcinoma, etc., since a cancer-related inflammatory microenvironment can be reflected in the blood as measurable parameters [21]. The basic changes are reported as neutrophilia, thrombocytosis, and lymphocytopenia [11–13]. Owing to challenges related with clinical adaptations of separate counts of them, ratios of these markers such as NLR and PLR are evaluated, and have been used as prognostic factors. There are also publications about predictive or prognostic values of NLR and PLR in endometrial premalignant lesions and EC [10–14]. Wang et al. found that stromal invasion in EC was significantly related with high values of both NLR and PLR values [22]. Haruma et al. reported this association for only PLR value [23]. An another recent study determined that elevated NLR was significantly associated with shorter survival in a univariate analysis, but no statistically significant relationship was found in multivariate analysis [11]. Cummings et al. showed that PLR and NLR were independent prognostic factors for EC with advanced stage [24]. However, Kurtoglu et al. reported that neither NLR nor PLR predicted stage or lymphovascular stromal invasion in EC [25].

Although the first studies have suggested the predictive or prognostic role of NLR and/or PLR values on a variety of carcinomas including EC, both NLR and PLR showed no statistical difference during malign transformation of endometrial mucosa in our patients. However, there was still a significant increase in the platelet levels of patients with atypia and EC (both, p<0.05). Moreover, thrombocytosis was most remarkable in the patients with advanced EC (p<0.01).

It has long been known that thrombocytosis is an adverse prognostic factor in many types of carcinomas including breast, renal cell, colorectal, lung and gynecologic [26, 27]. An increased platelet count has been shown to be associated with the risk of future cancers, as well [28]. Thrombocytosis is explained by the paraneoplastic phenomenon that arises from tumor

secretion of the proinflammatory cytokine interleukin-6, which increases thrombopoietin [29]. However, this mechanism is still not clear. On the other hand, anemia is a common finding in all patients with benign or malignant endometrial lesion and abnormal uterine bleeding. Therefore, an increase in hemoglobin and hematocrit values in our patients with early EC can be explained by tumor-induced secretion of erythropoietin, similar to thrombopoietin. It can also be prescribed as a drug for cancer- or chemotherapy-induced anemia, especially in advanced disease [30].

The major limitation of our study is its retrospective design, which may cause difficulties in controlling for potential confounding bias. However, a considerable number of patients with only epithelial endometrial cancer in a single center is evaluated in the present study.

In conclusion, our findings will cause questioning the accuracy of NLR and/or PLR' predictive value in EC, and regain the value of cancer-associated thrombocytosis.

## Author Contributions

**Conceptualization:** Aysun Firat, Fatma Ferda Verit Atmaca.

**Data curation:** Aysun Firat, Aysegul Ercan.

**Formal analysis:** Aysun Firat, Fatma Ferda Verit Atmaca.

**Investigation:** Aysun Firat, Aysegul Ercan, Fatma Ferda Verit Atmaca.

**Methodology:** Aysun Firat, Aysegul Ercan.

**Project administration:** Aysun Firat, Cengiz Mordeniz, Fatma Ferda Verit Atmaca.

**Software:** Aysun Firat.

**Supervision:** Aysun Firat.

**Validation:** Aysun Firat, Aysegul Ercan.

**Writing – original draft:** Aysun Firat, Aysegul Ercan.

**Writing – review & editing:** Aysun Firat, Cengiz Mordeniz, Fatma Ferda Verit Atmaca.

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
