## [Decision Letter · Decision Letter 0]

3 Nov 2022

PONE-D-22-22555Predictive value of hemogram parameters in malignant transformation of the endometrium in patients with different risk factorsPLOS ONE

Dear Dr. Authors,

Thank you for submitting your manuscript to PLOS ONE. After careful consideration, we feel that it has merit but does not fully meet PLOS ONE’s publication criteria as it currently stands. Therefore, we invite you to submit a revised version of the manuscript that addresses the points raised during the review process.

We look forward to receiving your revised manuscript.

Kind regards,

Tomasz Urbanowicz

Academic Editor

PLOS ONE

Journal Requirements:

a) Did participants provide their written or verbal informed consent to participate in this study?

Reviewers' comments:

Reviewer's Responses to Questions

**Comments to the Author**

1. Is the manuscript technically sound, and do the data support the conclusions?

Reviewer #1: Yes

Reviewer #2: Yes

2. Has the statistical analysis been performed appropriately and rigorously? 

Reviewer #1: Yes

Reviewer #2: Yes

3. Have the authors made all data underlying the findings in their manuscript fully available?

Reviewer #1: Yes

Reviewer #2: Yes

4. Is the manuscript presented in an intelligible fashion and written in standard English?

Reviewer #1: Yes

Reviewer #2: Yes

5. Review Comments to the Author

Reviewer #1: To Authors: The present study entitled as ‘Predictive value of hemogram parameters in malignant transformation of the endometrium in patients with different risk factors’ is a clinical research on one of the most controversial subjects in inflammation and endometrial carcinoma. The authors investigate neutrophil/ lymphocyte and platelet/lymphocyte ratios (NLR and PLR) in different stages of endometrial pathology in a range from polyps to atypia and further to the carcinoma sequence. On the contrary to several previous publications on endometrial cancer, you conclude that NLR and PLR values showed no statistical difference in your patients. However, your results still support the impact of inflammation on malign transformation from normal endometrial mucosa to atypia and carcinoma. The subject is of the last 10 years’ debatable topic among clinicians, since the recent publications from different areaa of expertises not always overlap with each other, and there is a scarce published study on endometrial carcinoma sequence. In general, the text is well-designed but there are some points listed below for revision;

- Abstract: Please re-write the sentence ‘NLR and PLR values showed no statistical difference in our patients.’ In conclusion section. The last three words seem to be unnecessary.

- Introduction: What do you mean in the last sentence of the last paragraph: ‘An easy, reproducible and simple marker is still needed to estimate the phases of endometrial pathologic lesions, and distinguish them from pathologically normal results’. Pathologically normal? Please re-write or correct.

- Introduction: There is no need for a subtitle in the last paragraph. Delete it.

- Methods: You need to extend the exclusion criteria. You excluded just patients with other inflammatory diseases? Be clear in defining this section.

- Methods: In the second paragraph, you state that high BMI is a risk factor, but you list normal BMI after parenthesis. Please correct.

- Methods: The abbreviation of SD should be written openly first. In the same paragraphy ‘…Group comparisons… group or groups?’. Please verify.

- Results: ‘…(Group II) (53.4 years vs 42.2 and 42 years; each, p<0.05)…’ and or vs? In the same section ‘The groups were compared by calculated NLR and PLR results, and there were no statistically meaningful relation…’ were or was?

- Discussion: In the first paragraph, ‘…menstruation, PCOS, HRT and tamoxifen use etc., and all can be viewed as factors increasing exposure of the endometrium to inflammation [2-4].’ Viewed? What you mean?

- Discussion: In the third paragraph, ‘…Haruma et al. reported this association for only PLR value [23]. An another recent study determined that…’ An another? What you mean?

- References: Please be compatible with the journal’s reference rules. PLOS uses the numbered citation (citation-sequence) method and first six authors, et al. Example: Hou WR, Hou YL, Wu GF, Song Y, Su XL, Sun B, et al. cDNA, genomic sequence cloning and overexpression of ribosomal protein gene L9 (rpL9) of the giant panda (Ailuropoda melanoleuca). Genet Mol Res. 2011;10: 1576-1588. Devaraju P, Gulati R, Antony PT, Mithun CB, Negi VS. Susceptibility to SLE in South Indian Tamils may be influenced by genetic selection pressure on TLR2 and TLR9 genes. Mol Immunol. 2014 Nov 22. pii: S0161-5890(14)00313-7. doi: 10.1016/j.molimm.2014.11.005. A DOI number for the full-text article is acceptable as an alternative to or in addition to traditional volume and page numbers. When providing a DOI, adhere to the format in the example above with both the label and full DOI included at the end of the reference (doi: 10.1016/j.molimm.2014.11.005). Do not provide a shortened DOI or the URL.

- Authors’ contrubution the study should be personalized such as Sho Matsubara, Data curation, Formal analysis, Writing – original draft, Seiji Mabuchi, Conceptualization, Data curation, Formal analysis, Writing – original draft,* Yoshinori Takeda, Data curation, Formal analysis, Naoki Kawahara, Data curation, Formal analysis, and Hiroshi Kobayashi, Supervision

XXXXXXX, Editor

Reviewer #2: This is a clinical study investigating the role of chronc inflammation in transition to endometrial cancer

since there are recent published works claiming neutrophil-to-lymphocyte ratio (NLR) and the platelet-to-

lymphocyte ratio (PLR) are important prognostic indicators in a variety of solid tumors. The construction

of the methodology including group allocations, sample size, power of statistics and the tables reflecting

the data are all logical and well presented. The purpose of the study and the results are compatible and

discussed in detail. Conclusions are drawn appropriately based on the data presented, as well. I believe

that this is an article with useful information and readable with interest. Language of writing is clear

except some minor punctiation errors which can be edited during publication process. Consequently, my

opinion is positive. This was a great honor for me to be selected as a reviewer by your journal.

6. PLOS authors have the option to publish the peer review history of their article (what does this mean?). If published, this will include your full peer review and any attached files.

Reviewer #1: **Yes: **Sema Yuksekdag, MD

Reviewer #2: No

---

## [Author Response · Author response to Decision Letter 0]

3 Nov 2022

Tomasz Urbanowicz

Academic Editor

PLOS ONE

Subject: Answers to the critics of Reviewers for the manuscript entitled as ‘PONE-D-22-22555, Predictive value of hemogr4m parameters in malignant transformation of the endometrium in patients with different risk factors’

Date: November 03, 2022

Dear Editor,

We received and answered the critics made by academic editor and reviewers. We revised the manusript according to the comments of reviewers and changed/added (new) sentences (in red color) concerning these critics. The manuscript is also edited for language faults as Reviewer 1 recommended, and all punctuation faults are corrected according to Reviewers 1 and 2. The revised final form is attached and sent through the journal’ s website. Looking forward to hearing from you soon. 

Sincerely.

Aysun Firat, MD, 

Specialist of Obstetris and Gynecology

To Academic Editor;

Response: We uploaded 3 files: Response to Reviewers (this letter), Revised Manuscript with Track Changes and Manuscript. 

a) Did participants provide their written or verbal informed consent to participate in this study?

Response: We added the sentence ‘Patients provided informed written consent to have data from their medical records used in research.’ (red colored, 1. paragraph, 1. sentence; before the sentence indicating the date and protocol number of Ethics’ Committee approval), since we always obtain informed consent forms signed by the patient at routine admission process.

3. In your Data Availability statement, you have not specified where the minimal data set underlying the results described in your manuscript can be found. PLOS defines a study's minimal data set as the underlying data used to reach the conclusions drawn in the manuscript and any additional data required to replicate the reported study findings in their entirety. All PLOS journals require that the minimal data set be made fully available. For more information about our data policy, please see http://journals.plos.org/plosone/s/data-availability. "Upon re-submitting your revised manuscript, please upload your study’s minimal underlying data set as either Supporting Information files or to a stable, public repository and include the relevant URLs, DOIs, or accession numbers within your revised cover letter. For a list of acceptable repositories, please see http://journals.plos.org/plosone/s/data-availability#loc-recommended-repositories. Any potentially identifying patient information must be fully anonymized. Important: If there are ethical or legal restrictions to sharing your data publicly, please explain these restrictions in detail. Please see our guidelines for more information on what we consider unacceptable restrictions to publicly sharing data: http://journals.plos.org/plosone/s/data-availability#loc-unacceptable-data-access-restrictions. Note that it is not acceptable for the authors to be the sole named individuals responsible for ensuring data access. We will update your Data Availability statement to reflect the information you provide in your cover letter.

Response: Data from this study are available upon request since there are legal restrictions (by Turkish Ministry of Health) on sharing data publicly. However, we anonymized the patients’ identities and protocol numbers on the system and saved whole data in Excel form. Data protection process and audit are supplied by Local Ethics’ Committe (ieahetikkurul@gmail.com, https://istanbuleah.saglik.gov.tr/TR-168206/kurul-uyeleri.html). The first and the corresponding author of the study (Aysun Firat, MD) may send them by email (aysunfiratsbuieah@gmail.com) on request. There is no change to our financial disclosure, because we have no any financial support for the study.

Response: Title page is revised, there was no problem.

Response: It is stated only in Methods section of the main text.

Response: References were revised according to the critics of Reviewer 1 and the rules of the journal. All were formatted accordingly. Thank you very much for your critics and support. Regards.

To Reviewer 1;

The present study entitled as ‘Predictive value of hemogram parameters in malignant transformation of the endometrium in patients with different risk factors’ is a clinical research on one of the most controversial subjects in inflammation and endometrial carcinoma. The authors investigate neutrophil/ lymphocyte and platelet/lymphocyte ratios (NLR and PLR) in different stages of endometrial pathology in a range from polyps to atypia and further to the carcinoma sequence. On the contrary to several previous publications on endometrial cancer, you conclude that NLR and PLR values showed no statistical difference in your patients. However, your results still support the impact of inflammation on malign transformation from normal endometrial mucosa to atypia and carcinoma. The subject is of the last 10 years’ debatable topic among clinicians, since the recent publications from different areaa of expertises not always overlap with each other, and there is a scarce published study on endometrial carcinoma sequence. In general, the text is well-designed but there are some points listed below for revision;

- Abstract: Please re-write the sentence ‘NLR and PLR values showed no statistical difference in our patients.’ In conclusion section. The last three words seem to be unnecessary.

Response: In conclusion section of Abstract, we re-wrote the sentence as ‘NLR and PLR values showed no statistical difference.’ (in red color, conclusion, Abstract).

- Introduction: What do you mean in the last sentence of the last paragraph: ‘An easy, reproducible and simple marker is still needed to estimate the phases of endometrial pathologic lesions, and distinguish them from pathologically normal results’. Pathologically normal? Please re-write or correct.

Response: We would like to thank to Reviewer for this important criticism. We changed the sentence as ‘An easy, reproducible and simple marker is still needed to predict the transition to invasive carcinoma, and distinguish this from the benign lesions.’

- Introduction: There is no need for a subtitle in the last paragraph. Delete it.

Response: It is deleted from the Introduction section.

- Methods: You need to extend the exclusion criteria. You excluded just patients with other inflammatory diseases? Be clear in defining this section.

Response: We changed the sentence as ‘Patients with missing information in their files and those with any documented infectious or inflammatory disorders were excluded.’(in red color, last sentence in Methods section, main text)

- Methods: In the second paragraph, you state that high BMI is a risk factor, but you list normal BMI after parenthesis. Please correct.

Response: We deleted the normal scores, and changed as ‘….. high body mass index (>25),…..’ (second paragraph, Methods, main text). 

- Methods: The abbreviation of SD should be written openly first. In the same paragraphy ‘…Group comparisons… group or groups?’. Please verify.

Response: The last paragraph of Methods section was changed as; ‘Statistical package for social sciences (SPSS version 11.5) was used for the statistical analysis. Descriptive values were expressed as number (n), %, median or mean with standard deviation (SD). Chi-square, Student’s t and Mann-Whitney U tests were used for nominal and categorical values, and Kruskal–Wallis test was used to compare the nonparametric variables. Group comparisons were done with one-way ANOVA and Tukey HSD tests to ascertain the group that cause the difference. P<0.05 was considered statistically significant.’

- Results: ‘…(Group II) (53.4 years vs 42.2 and 42 years; each, p<0.05)…’ and or vs? In the same section ‘The groups were compared by calculated NLR and PLR results, and there were no statistically meaningful relation…’ were or was?

Response: The suggested changes were made and marked in red color (2. And 3. Paragraphs of Results section, main text). 

- Discussion: In the first paragraph, ‘…menstruation, PCOS, HRT and tamoxifen use etc., and all can be viewed as factors increasing exposure of the endometrium to inflammation [2-4].’ Viewed? What you mean?

Response: We deleted this word (viewed); instead we used the phrase ‘….all are known as….’ (in red color, first paragraph of Discussion section, main text). 

- Discussion: In the third paragraph, ‘…Haruma et al. reported this association for only PLR value [23]. An another recent study determined that…’ An another? What you mean?

Response: We omitted ‘an’ (3. Paragraph, Disussion section of the main text).

- References: Please be compatible with the journal’s reference rules. PLOS uses the numbered citation (citation-sequence) method and first six authors, et al. Example: Hou WR, Hou YL, Wu GF, Song Y, Su XL, Sun B, et al. cDNA, genomic sequence cloning and overexpression of ribosomal protein gene L9 (rpL9) of the giant panda (Ailuropoda melanoleuca). Genet Mol Res. 2011;10: 1576-1588. Devaraju P, Gulati R, Antony PT, Mithun CB, Negi VS. Susceptibility to SLE in South Indian Tamils may be influenced by genetic selection pressure on TLR2 and TLR9 genes. Mol Immunol. 2014 Nov 22. pii: S0161-5890(14)00313-7. doi: 10.1016/j.molimm.2014.11.005. A DOI number for the full-text article is acceptable as an alternative to or in addition to traditional volume and page numbers. When providing a DOI, adhere to the format in the example above with both the label and full DOI included at the end of the reference (doi: 10.1016/j.molimm.2014.11.005). Do not provide a shortened DOI or the URL.

Response: We revised and formatted all references according to the journal ‘s rules. 

- Authors’ contrubution the study should be personalized such as Sho Matsubara, Data curation, Formal analysis, Writing – original draft, Seiji Mabuchi, Conceptualization, Data curation, Formal analysis, Writing – original draft,* Yoshinori Takeda, Data curation, Formal analysis, Naoki Kawahara, Data curation, Formal analysis, and Hiroshi Kobayashi, Supervision

XXXXXXX, Editor

Response to Reviewer 2;

This is a clinical study investigating the role of chronc inflammation in transition to endometrial cancer since there are recent published works claiming neutrophil-to-lymphocyte ratio (NLR) and the platelet-to-lymphocyte ratio (PLR) are important prognostic indicators in a variety of solid tumors. The construction of the methodology including group allocations, sample size, power of statistics and the tables reflecting the data are all logical and well presented. The purpose of the study and the results are compatible and discussed in detail. Conclusions are drawn appropriately based on the data presented, as well. I believe that this is an article with useful information and readable with interest. Language of writing is clear except some minor punctiation errors which can be edited during publication process. Consequently, my

opinion is positive. This was a great honor for me to be selected as a reviewer by your journal.

Response: Considering the critics of Reviewer, we made the changes throughout the text. Whole text was reviewed again for language and punctuation faults (red in color, throughout the text). 

We would like to express our special thanks to Reviewers for their contribution to our knowledge and also to this article. Regards.

---

## [Editor Report · Decision Letter 1]

14 Nov 2022

PONE-D-22-22555R1Predictive value of hemogram parameters in malignant transformation of the endometrium in patients with different risk factorsPLOS ONE

Dear Authors,

Thank you for submitting your manuscript to PLOS ONE. After careful consideration, we feel that it has merit but does not fully meet PLOS ONE’s publication criteria as it currently stands. Therefore, we invite you to submit a revised version of the manuscript that addresses the points raised during the review process.

 We look forward to receiving your revised manuscript.

Kind regards,

Tomasz Urbanowicz

Academic Editor

PLOS ONE

---

## [Author Response · Author response to Decision Letter 1]

16 Nov 2022

PONE-D-22-22555R2

Predictive value of hemogram parameters in malignant transformation of the endometrium in patients with different risk factors

Dr. Aysun Firat

Dear Dr. Firat,

We've checked your submission and before we can proceed, we need you to address the following issues:

1. We notice that your manuscript file was uploaded on Nov 3, 2022. Please can you upload the latest version of your revised manuscript as the main article file, ensuring that does not contain any tracked changes or highlighting. This will be used in the production process if your manuscript is accepted. Please follow this link for more information: http://blogs.PLOS.org/everyone/2011/05/10/how-to-submit-your-revised-manuscript/

We've returned your manuscript to your account. Please resolve these issues and resubmit your manuscript within 21 days. If you need more time, please email the journal office at plosone@plos.org. We are happy to grant extensions of up to one month past this due date. If we do not hear from you within 21 days, we will withdraw your manuscript.

Please log on to PLOS Editorial Manager at https://www.editorialmanager.com/pone/ to access your manuscript. You will find your manuscript in the 'Submissions Sent Back to Author' link under the New Submissions menu. Be sure to remove your previous manuscript file if you are uploading a new file in response to these requests. After you've made the changes requested above, please be sure to view and approve the revised PDF after rebuilding the PDF to complete the resubmission process.

We are requesting these changes to comply with the PLOS ONE submission guidelines (https://journals.plos.org/plosone/s/submission-guidelines). Please note that we won't send your manuscript for review until you have resolved the above requests.

Thank you for submitting your work to PLOS ONE and supporting our mission of Open Science.

Kind regards,

Jeunarine Repe Flores

---

## [Editor Report · Decision Letter 2]

5 Dec 2022

Predictive value of hemogram parameters in malignant transformation of the endometrium in patients with different risk factors

PONE-D-22-22555R2

Dear Authors,

We’re pleased to inform you that your manuscript has been judged scientifically suitable for publication and will be formally accepted for publication once it meets all outstanding technical requirements.

Kind regards,

Tomasz Urbanowicz

Academic Editor

PLOS ONE

---

## [Editor Report · Acceptance letter]

3 Jan 2023

PONE-D-22-22555R2 

Predictive value of hemogram parameters in malignant transformation of the endometrium in patients with different risk factors 

Dear Dr. Firat:

I'm pleased to inform you that your manuscript has been deemed suitable for publication in PLOS ONE. Congratulations! Your manuscript is now with our production department. 

Kind regards, 

on behalf of

MR Tomasz Urbanowicz 

Academic Editor

PLOS ONE